# Neural Non-Rigid Tracking

**Aljaž Božič**[1]*
aljaz.bozic@tum.de

**Pablo Palafox**[1]*
pablo.palafox@tum.de

**Michael Zollhöfer**[2]    **Angela Dai**[1]    **Justus Thies**[1]    **Matthias Nießner**[1]

[1]Technical University of Munich          [2]Stanford University

## Abstract

We introduce a novel, end-to-end learnable, differentiable non-rigid tracker that enables state-of-the-art non-rigid reconstruction by a learned robust optimization. Given two input RGB-D frames of a non-rigidly moving object, we employ a convolutional neural network to predict dense correspondences and their confidences. These correspondences are used as constraints in an as-rigid-as-possible (ARAP) optimization problem. By enabling gradient back-propagation through the weighted non-linear least squares solver, we are able to learn correspondences and confidences in an end-to-end manner such that they are optimal for the task of non-rigid tracking. Under this formulation, correspondence confidences can be learned via self-supervision, informing a learned robust optimization, where outliers and wrong correspondences are automatically down-weighted to enable effective tracking. Compared to state-of-the-art approaches, our algorithm shows improved reconstruction performance, while simultaneously achieving $85\times$ faster correspondence prediction than comparable deep-learning based methods. We make our code available at `https://github.com/DeformableFriends/NeuralTracking`.

## 1   Introduction

The capture and reconstruction of real-world environments is a core problem in computer vision, enabling numerous VR/AR applications. While there has been significant progress in reconstructing static scenes, tracking and reconstruction of dynamic objects remains a challenge. Non-rigid reconstruction focuses on dynamic objects, without assuming any explicit shape priors, such as human or face parametric models. Commodity RGB-D sensors, such as Microsoft's Kinect or Intel's Realsense, provide a cost-effective way to acquire both color and depth video of dynamic motion. Using a large number of RGB-D sensors can lead to an accurate non-rigid reconstruction, as shown by Dou et al. [8]. Our work focuses on non-rigid reconstruction from a single RGB-D camera, thus eliminating the need for specialized multi-camera setups.

The seminal DynamicFusion by Newcombe et al. [23] introduced a non-rigid tracking and mapping pipeline that uses depth input for real-time non-rigid reconstruction from a single RGB-D camera. Various approaches have expanded upon this framework by incorporating sparse color correspondences [13] or dense photometric optimization [10]. DeepDeform [4] presented a learned correspondence prediction, enabling significantly more robust tracking of fast motion and re-localization. Unfortunately, the computational cost of the correspondence prediction network ($\sim 2$ seconds per frame for a relatively small number of non-rigid correspondences) inhibits real-time performance.

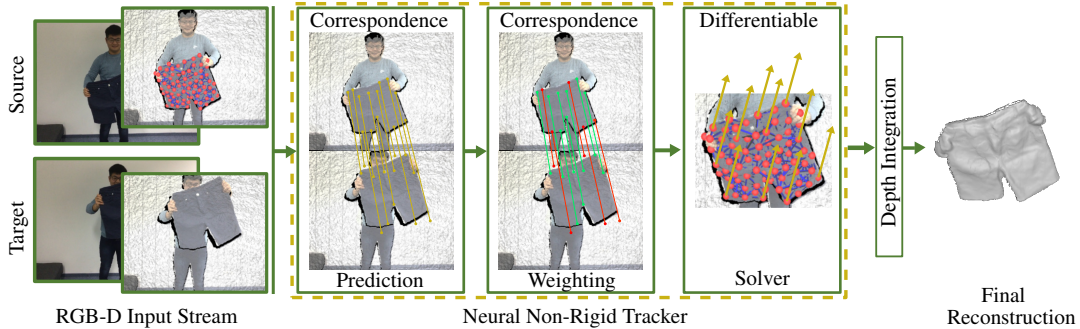

Figure 1: Neural Non-Rigid Tracking: based on RGB-D input data of a source and a target frame, our learned non-rigid tracker estimates the non-rigid deformations to align the source to the target frame. We propose an end-to-end approach, enabling correspondences and their importance weights to be informed by the non-rigid solver. Similar to robust optimization, this provides robust tracking, and the resulting deformation field can then be used to integrate the depth observations in a canonical volumetric 3D grid that implicitly represents the surface of the object (final reconstruction).

Simultaneously, work on learned optical flow has shown dense correspondence prediction at real-time rates [30]. However, directly replacing the non-rigid correspondence predictions from Božič et al. [4] with these optical flow predictions does not produce accurate enough correspondences for comparable non-rigid reconstruction performance. In our work, we propose a neural non-rigid tracker, i.e., an end-to-end differentiable non-rigid tracking pipeline which combines the advantages of classical deformation-graph-based reconstruction pipelines [23, 13] with novel learned components. Our end-to-end approach enables learning outlier rejection in a self-supervised manner, which guides a robust optimization to mitigate the effect of inaccurate correspondences or major occlusions present in single RGB-D camera scenarios.

Specifically, we cast the non-rigid tracking problem as an as-rigid-as-possible (ARAP) optimization problem, defined on correspondences between points in a source and a target frame. A differentiable Gauss-Newton solver allows us to obtain gradients that enable training a neural network to predict an importance weight for every correspondence in a completely self-supervised manner, similar to robust optimization. The end-to-end training significantly improves non-rigid tracking performance. Using our neural tracker in a non-rigid reconstruction framework results in $85\times$ faster correspondence prediction and improved reconstruction performance compared to the state of the art.

In summary, we propose a novel neural non-rigid tracking approach with two key contributions:

- an end-to-end differentiable Gauss-Newton solver, which provides gradients to better inform a correspondence prediction network used for non-rigid tracking of two frames;

- a self-supervised approach for learned correspondence weighting, which is informed by our differentiable solver and enables efficient, robust outlier rejection, thus, improving non-rigid reconstruction performance compared to the state of the art.

## 2   Related Work

**Non-rigid Reconstruction.**   Reconstruction of deformable objects using a single RGB-D camera is an important research area in computer vision. State-of-the-art methods rely on deformation graphs [29, 35] that enable robust and temporally consistent 3D motion estimation. While earlier approaches required an object template, such graph-based tracking has been extended to simultaneous tracking and reconstruction approaches [7, 23]. These works used depth fitting optimization objectives in the form of iterative closest points, or continuous depth fitting in [26, 27]. Rather than relying solely on depth information, recent works have incorporated SIFT features [13], dense photometric fitting [10], and sparse learned correspondence [4].

**Correspondence Prediction for Non-rigid Tracking.** In non-rigid tracking, correspondences must be established between the two frames we want to align. While methods such as DynamicFusion [23] rely on projective correspondences, recent methods leverage learned correspondences [4]. DeepDeform [4] relies on sparse predicted correspondences, trained on an annotated dataset of deforming objects. Since prediction is done independently for each correspondence, this results in a high compute cost, compared to dense predictions of state-of-the-art optical flow networks. Optical flow [6, 12, 30, 18] and scene flow [21, 3, 19, 33] methods achieve promising results in predicting dense correspondences between two frames, with some approaches not even requiring direct supervision [32, 16, 15]. In our proposed neural non-rigid tracking approach, we build upon PWC-Net [30] for dense correspondence prediction to inform our non-rigid deformation energy formulation. Since our approach allows for end-to-end training, our 2D correspondence prediction finds correspondences better suited for non-rigid tracking.

**Differentiable Optimization.** Differentiable optimizers have been explored for various tasks, including image alignment [5], rigid pose estimation [11, 20], multi-frame direct bundle-adjustment [31], and rigid scan-to-CAD alignment [1]. In addition to achieving higher accuracy, an end-to-end differentiable optimization approach also offers the possibility to optimize run-time, as demonstrated by learning efficient pre-conditioning methods in [9, 25, 17]. Unlike Li et al. [17], which employs an image-based tracker (with descriptors defined on nodes in a pixel-aligned graph), our approach works on general graphs and learns to robustify correspondence prediction for non-rigid tracking by learning self-supervised correspondence confidences.

# 3 Non-Rigid Reconstruction Notation

Non-rigid alignment is a crucial part of non-rigid reconstruction pipelines. In the single RGB-D camera setup, we are given a pair of source and target RGB-D frames $\{(\mathcal{I}_s, \mathcal{P}_s), (\mathcal{I}_t, \mathcal{P}_t)\}$, where $\mathcal{I}_* \in \mathbb{R}^{H \times W \times 3}$ is an RGB image and $\mathcal{P}_* \in \mathbb{R}^{H \times W \times 3}$ a 3D point image. The goal is to estimate a warp field $\mathcal{Q} : \mathbb{R}^3 \mapsto \mathbb{R}^3$ that transforms $\mathcal{P}_s$ into the target frame. Note that we define the 3D point image $\mathcal{P}_s$ as the result of back-projecting every pixel $\mathbf{u} \in \Pi_s \subset \mathbb{R}^2$ into the camera coordinate system with given camera intrinsic parameters. To this end, we define the inverse of the perspective projection to back-project a pixel $\mathbf{u}$ given the pixel's depth $d_\mathbf{u}$ and the intrinsic camera parameters $\mathbf{c}$:

$$\pi_\mathbf{c}^{-1} : \mathbb{R}^2 \times \mathbb{R} \to \mathbb{R}^3, \quad (\mathbf{u}, d_\mathbf{u}) \mapsto \pi_\mathbf{c}^{-1}(\mathbf{u}, d_\mathbf{u}) = \mathbf{p}. \tag{1}$$

To maintain robustness against noise in the depth maps, state-of-the-art approaches define an embedded deformation graph $\mathcal{G} = \{\mathcal{V}, \mathcal{E}\}$ over the *source* RGB-D frame, where $\mathcal{V}$ is the set of graph nodes defined by their 3D coordinates $\mathbf{v}_i \in \mathbb{R}^3$ and $\mathcal{E}$ the set of edges between nodes, as described in [29] and illustrated in Fig. 1. Thus, for every node in $\mathcal{G}$, a global translation vector $\mathbf{t}_{\mathbf{v}_i} \in \mathbb{R}^3$ and a rotation matrix $\mathbf{R}_{\mathbf{v}_i} \in \mathbb{R}^{3 \times 3}$, must be estimated in the alignment process. We parameterize rotations with a 3-dimensional axis-angle vector $\boldsymbol{\omega} \in \mathbb{R}^3$. We use the exponential map $\exp : \mathfrak{so}(3) \to \mathrm{SO}(3), \quad \widehat{\boldsymbol{\omega}} \mapsto e^{\widehat{\boldsymbol{\omega}}} = \mathbf{R}$ to convert from axis-angle to matrix rotation form, where the $\widehat{\cdot}$-operator creates a $3 \times 3$ skew-symmetric matrix from a 3-dimensional vector. The resulting graph motion is denoted by $\mathcal{T} = (\boldsymbol{\omega}_{\mathbf{v}_1}, \mathbf{t}_{\mathbf{v}_1}, \ldots, \boldsymbol{\omega}_{\mathbf{v}_N}, \mathbf{t}_{\mathbf{v}_N}) \in \mathbb{R}^{N \times 6}$ for a graph with $N$ nodes.

Dense motion can then be computed by interpolating the nodes' motion $\mathcal{T}$ by means of a warping function Q. When applied to a 3D point $\mathbf{p} \in \mathbb{R}^3$, it produces the point's deformed position

$$Q(\mathbf{p}, \mathcal{T}) = \sum_{\mathbf{v}_i \in \mathcal{V}} \alpha_{\mathbf{v}_i}(e^{\widehat{\boldsymbol{\omega}}_{\mathbf{v}_i}}(\mathbf{p} - \mathbf{v}_i) + \mathbf{v}_i + \mathbf{t}_{\mathbf{v}_i}). \tag{2}$$

The weights $\alpha_{\mathbf{v}_i} \in \mathbb{R}$, also known as *skinning* weights, measure the influence of each node on the current point $\mathbf{p}$ and are computed as in [34]. Please see the supplemental material for further detail.

# 4 Neural Non-rigid Tracking

Given a pair of source and target RGB-D frames $(\mathcal{Z}_s, \mathcal{Z}_t)$, where $\mathcal{Z}_* = (\mathcal{I}_* | \mathcal{P}_*) \in \mathbb{R}^{H \times W \times 6}$ is the concatenation of an RGB and a 3D point image as defined in Section 3, we aim to find a function $\Theta$ that estimates the motion $\mathcal{T}$ of a deformation graph $\mathcal{G}$ with $N$ nodes (given by their 3D coordinates $\mathcal{V}$) defined over the source RGB-D frame. This implicitly defines source-to-target dense 3D motion (see Figure 2). Formally, we have:

$$\Theta : \mathbb{R}^{H \times W \times 6} \times \mathbb{R}^{H \times W \times 6} \times \mathbb{R}^{N \times 3} \to \mathbb{R}^{N \times 6}, \quad (\mathcal{Z}_s, \mathcal{Z}_t, \mathcal{V}) \mapsto \Theta(\mathcal{Z}_s, \mathcal{Z}_t, \mathcal{V}) = \mathcal{T}. \tag{3}$$

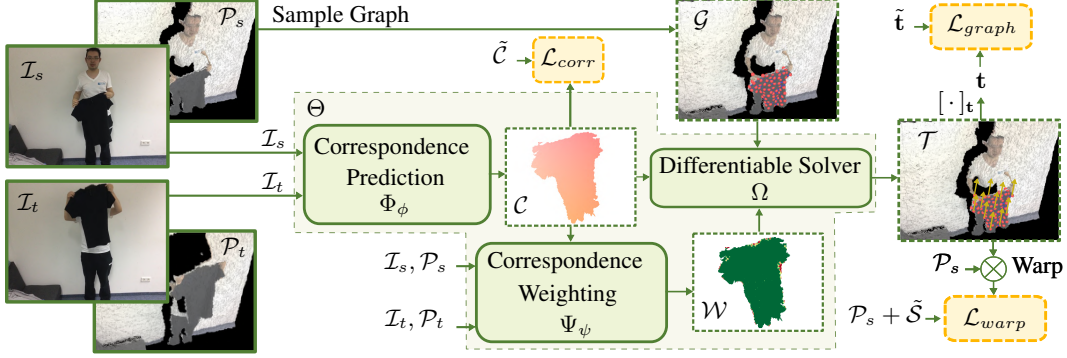

Figure 2: Overview of our neural non-rigid tracker. Given a pair of source and target images, $\mathcal{I}_s$ and $\mathcal{I}_t$, a dense correspondence map $\mathcal{C}$ between the frames is estimated via a convolutional neural network $\Phi$. Importance weights $\mathcal{W}$ for these correspondences are computed through a function $\Psi$. Together with a graph $\mathcal{G}$ defined over the source RGB-D frame $\mathcal{P}_s$, both $\mathcal{C}$ and $\mathcal{W}$ are input to a differentiable solver $\Omega$. The solver outputs the graph motion $\mathcal{T}$, i.e., the non-rigid alignment between source and target frames. Our approach is optimized end-to-end, with losses on the final alignment using $\mathcal{L}_{\text{graph}}$ and $\mathcal{L}_{\text{warp}}$, and an intermediate loss on the correspondence map $\mathcal{L}_{\text{corr}}$.

To estimate $\mathcal{T}$, we first establish dense 2D correspondences between the source and target frame using a deep neural network $\Phi$. These correspondences, denoted as $\mathcal{C}$, are used to construct the data term in our non-rigid alignment optimization. Since the presence of outlier correspondence predictions has a strong negative impact on the performance of non-rigid tracking, we introduce a weighting function $\Psi$, inspired by robust optimization, to down-weight inaccurate predictions. Function $\Psi$ outputs importance weights $\mathcal{W}$ and is learned in a self-supervised manner. Finally, both correspondence predictions $\mathcal{C}$ and importance weights $\mathcal{W}$ are input to a differentiable, non-rigid alignment optimization module $\Omega$. By optimizing the non-rigid alignment energy (see Section 4.3), the differentiable optimizer $\Omega$ estimates the deformation graph parameters $\mathcal{T}$ that define the motion from source to target frame:

$$\mathcal{T} = \Theta\left(\mathcal{Z}_s, \mathcal{Z}_t, \mathcal{V}\right) = \Omega\left(\Phi(\cdot), \Psi(\cdot), \mathcal{V}\right) = \Omega\left(\mathcal{C}, \mathcal{W}, \mathcal{V}\right). \tag{4}$$

In the following, we define the dense correspondence predictor $\Phi$, the importance weighting $\Psi$ and the optimizer $\Omega$, and describe a fully differentiable approach for optimizing $\Phi$ and $\Psi$ such that we can estimate dense correspondences with importance weights best suited for non-rigid tracking.

### 4.1 Dense Correspondence Prediction

The dense correspondence prediction function $\Phi$ takes as input a pair of source and target RGB images $(\mathcal{I}_s, \mathcal{I}_t)$, and for each source pixel location $\mathbf{u} \in \Pi_s \subset \mathbb{R}^2$ it outputs a corresponding pixel location in the target image $\mathcal{I}_t$, which we denote by $\mathbf{c_u} \in \Pi_t \subset \mathbb{R}^2$. Formally, $\Phi$ is defined as

$$\Phi : \mathbb{R}^{H \times W \times 3} \times \mathbb{R}^{H \times W \times 3} \to \mathbb{R}^{H \times W \times 2}, \quad (\mathcal{I}_s, \mathcal{I}_t) \mapsto \Phi\left(\mathcal{I}_s, \mathcal{I}_t\right) = \mathcal{C}, \tag{5}$$

where $\mathcal{C}$ is the resulting dense correspondence map. The function $\Phi$ is represented by a deep neural network that leverages the architecture of a state-of-the-art optical flow estimator [30].

### 4.2 Correspondence Importance Weights

For each source pixel $\mathbf{u} \in \Pi_s \subset \mathbb{R}^2$ and its correspondence $\mathbf{c_u} \in \Pi_t \subset \mathbb{R}^2$, we additionally predict an importance weight $w_{\mathbf{u}} \in (0, 1)$ by means of the weighting function $\Psi$. The latter takes as input the source RGB-D image $\mathcal{Z}_s$, the corresponding sampled target frame values $\mathcal{Z}'_t$, and intermediate features from the correspondence network $\Phi$, and outputs weights for the correspondences between source and target. Note that $\mathcal{Z}'_t$ is the result of bilinearly sampling [14] the target image $\mathcal{Z}_t$ at the predicted correspondence locations $\mathcal{C}$. The last layer of features $\mathcal{H}$ of the correspondence network $\Phi$, with dimension $D = 565$, are used to inform $\Psi$. The weighting function is thus defined as

$$\Psi : \mathbb{R}^{H \times W \times 6} \times \mathbb{R}^{H \times W \times 6} \times \mathbb{R}^{H \times W \times D} \to \mathbb{R}^{H \times W \times 1}, \quad (\mathcal{Z}_s, \mathcal{Z}'_t, \mathcal{H}) \mapsto \Psi\left(\mathcal{Z}_s, \mathcal{Z}'_t, \mathcal{H}\right) = \mathcal{W}. \tag{6}$$

## 4.3 Differentiable Optimizer

We introduce a differentiable optimizer $\Omega$ to estimate the deformation graph parameters $\mathcal{T}$, given the correspondence map $\mathcal{C}$, importance weights $\mathcal{W}$, and $N$ graph nodes $\mathcal{V}$:

$$\Omega : \mathbb{R}^{H \times W \times 2} \times \mathbb{R}^{H \times W \times 1} \times \mathbb{R}^{N \times 3} \to \mathbb{R}^{N \times 6}, \quad (\mathcal{C}, \mathcal{W}, \mathcal{V}) \mapsto \Omega(\mathcal{C}, \mathcal{W}, \mathcal{V}) = \mathcal{T}, \quad (7)$$

with $\mathcal{C}$ and $\mathcal{W}$ estimated by functions $\Phi$ (Eq. 5) and $\Psi$ (Eq. 6), respectively. Using the predicted dense correspondence map $\mathcal{C}$, we establish the data term for the non-rigid tracking optimization. Specifically, we use a 2D data term that operates in image space and a depth data term that leverages the depth information of the input frames. In addition to the data terms, we employ an As-Rigid-As-Possible regularizer [28] to encourage node deformations to be locally rigid, enabling robust deformation estimates even in the presence of noisy input cues. Note that the resulting optimizer module $\Omega$ is fully differentiable, but contains no learnable parameters. In summary, we formulate non-rigid tracking as the following nonlinear optimization problem:

$$\arg\min_{\mathcal{T}} \left( \lambda_{\text{2D}} E_{\text{2D}}(\mathcal{T}) + \lambda_{\text{depth}} E_{\text{depth}}(\mathcal{T}) + \lambda_{\text{reg}} E_{\text{reg}}(\mathcal{T}) \right). \quad (8)$$

**2D reprojection term.** Given the outputs of the dense correspondence predictor and weighting function, $\Phi(\mathcal{I}_s, \mathcal{I}_t)$ and $\Psi(\mathcal{Z}_s, \mathcal{Z}'_t, \mathcal{H})$, respectively, we query for every pixel $\mathbf{u}$ in the source frame its correspondence $\mathbf{c_u}$ and weight $w_{\mathbf{u}}$ to build the following energy term:

$$E_{\text{2D}}(\mathcal{T}) = \sum_{\mathbf{u} \in \Pi_s} w_{\mathbf{u}}^2 \left\| \pi_{\mathbf{c}}(Q(\mathbf{p_u}, \mathcal{T})) - \mathbf{c_u} \right\|_2^2, \quad (9)$$

where $\pi_{\mathbf{c}} : \mathbb{R}^3 \to \mathbb{R}^2, \quad \mathbf{p} \mapsto \pi_{\mathbf{c}}(\mathbf{p})$ is a perspective projection with intrinsic parameters $\mathbf{c}$ and $\mathbf{p_u} = \pi_{\mathbf{c}}^{-1}(\mathbf{u}, d_{\mathbf{u}})$ as defined in Eq. 1. Each pixel is back-projected to 3D, deformed using the current graph motion estimate as described in Eq. 2 and projected onto the target image plane. The projected deformed location is compared to the predicted correspondence $\mathbf{c_u}$.

**Depth term.** The depth term leverages the depth cues of the source and target images. Specifically, it compares the $z$ components of a warped source point, i.e., $[Q(\mathbf{p_u}, \mathcal{T})]_z$, and a target point sampled at the corresponding location $\mathbf{c_u}$ using bilinear interpolation:

$$E_{\text{depth}}(\mathcal{T}) = \sum_{\mathbf{u} \in \Pi_s} w_{\mathbf{u}}^2 \left( [Q(\mathbf{p_u}, \mathcal{T})]_z - [P_t(\mathbf{c_u})]_z \right)^2. \quad (10)$$

**Regularization term.** We encourage the deformation of neighboring nodes in the deformation graph to be locally rigid. Each node $\mathbf{v}_i \in \mathcal{V}$ has at most $K = 8$ neighbors in the set of edges $\mathcal{E}$, computed as nearest nodes using geodesic distances. The regularization term follows [28]:

$$E_{\text{reg}}(\mathcal{T}) = \sum_{(\mathbf{v}_i, \mathbf{v}_j) \in \mathcal{E}} \left\| e^{\widehat{\boldsymbol{\omega}}_{\mathbf{v}_i}} (\mathbf{v}_j - \mathbf{v}_i) + \mathbf{v}_i + \mathbf{t}_{\mathbf{v}_i} - (\mathbf{v}_j + \mathbf{t}_{\mathbf{v}_j}) \right\|_2^2. \quad (11)$$

Equation 8 is minimized using the Gauss-Newton algorithm, as described in Algorithm 1. In the following, we denote the number of correspondences by $|\mathcal{C}|$ and the number of graph edges by $|\mathcal{E}|$. Moreover, we transform all energy terms into a residual vector $\mathbf{r} \in \mathbb{R}^{3|\mathcal{C}|+3|\mathcal{E}|}$. For every graph node, we compute partial derivatives with respect to translation and rotation parameters, constructing a Jacobian matrix $\mathbf{J} \in \mathbb{R}^{(3|\mathcal{C}|+3|\mathcal{E}|) \times 6N}$, where $N$ is the number of nodes in the set of vertices $\mathcal{V}$. Analytic formulas for partial derivatives are described in the supplemental material.

Initially, the deformation parameters are initialized to $\mathcal{T}_0 = \mathbf{0}$, corresponding to zero translation and identity rotations. In each iteration $n$, the residual vector $\mathbf{r}_n$ and the Jacobian matrix $\mathbf{J}_n$ are computed using the current estimate $\mathcal{T}_n$, and the following linear system is solved (using LU decomposition) to compute an increment $\Delta\mathcal{T}$:

$$\mathbf{J}_n^T \mathbf{J}_n \Delta\mathcal{T} = -\mathbf{J}_n^T \mathbf{r}_n. \quad (12)$$

At the end of every iteration, the motion estimate $\mathcal{T}$ is updated as $\mathcal{T}_{n+1} = \mathcal{T}_n + \Delta\mathcal{T}$. Most operations are matrix-matrix or matrix-vector multiplications, which are trivially differentiable. Derivatives of the linear system solve operation are computed analytically, as described in [2] and detailed in the supplement. We use $max\_iter = 3$ Gauss-Newton iterations, which encourages the correspondence prediction and weight functions, $\Phi$ and $\Psi$, respectively, to make predictions such that convergence in 3 iterations is possible. In our experiments we use $(\lambda_{\text{2D}}, \lambda_{\text{depth}}, \lambda_{\text{reg}}) = (0.001, 1, 1)$.

---

**Algorithm 1** Gauss-Newton Optimization

---

1: $\mathcal{C} \leftarrow \Phi\left(\mathcal{I}_s, \mathcal{I}_t\right)$                                                         ▷ Estimate correspondences
2: $\mathcal{W} \leftarrow \Psi\left(\mathcal{Z}_s, \mathcal{Z}'_t, \mathcal{H}\right)$                                          ▷ Estimate importance weights
3: **function** SOLVER($\mathcal{C}, \mathcal{W}, \mathcal{V}$)
4:      $\mathcal{T} \leftarrow \mathbf{0}$
5:      **for** $n \leftarrow 0$ to $max\_iter$ **do**
6:          $\mathbf{J}, \mathbf{r} \leftarrow$ ComputeJacobianAndResidual($\mathcal{V}, \mathcal{T}, \mathcal{Z}_s, \mathcal{Z}'_t, \mathcal{C}, \mathcal{W}$)
7:          $\Delta\mathcal{T} \leftarrow$ LUDecomposition($\mathbf{J}^T\mathbf{J}\Delta\mathcal{T} = -\mathbf{J}^T\mathbf{r}$)           ▷ Solve linear system
8:          $\mathcal{T} \leftarrow \mathcal{T} + \Delta\mathcal{T}$                              ▷ Apply increment
9:      **return** $\mathcal{T}$

---

### 4.4 End-to-end Optimization

Given a dataset of samples $\mathcal{X}_{s,t} = \{[\mathcal{I}_s|\mathcal{P}_s], [\mathcal{I}_t|\mathcal{P}_t], \mathcal{V}\}$, our goal is to find the parameters $\phi$ and $\psi$ of $\Phi_\phi$ and $\Psi_\psi$, respectively, so as to estimate the motion $\mathcal{T}$ of a deformation graph $\mathcal{G}$ defined over the source RGB-D frame. This can be formulated as a differentiable optimization problem (allowing for back-propagation) with the following objective:

$$\underset{\phi,\psi}{\arg\min} \sum_{\mathcal{X}_{s,t}} \lambda_{\text{corr}}\mathcal{L}_{\text{corr}}(\phi) + \lambda_{\text{graph}}\mathcal{L}_{\text{graph}}(\phi,\psi) + \lambda_{\text{warp}}\mathcal{L}_{\text{warp}}(\phi,\psi) \tag{13}$$

**Correspondence loss.** We use a robust $q$-norm as in [30] to enforce closeness of correspondence predictions to ground-truth:

$$\mathcal{L}_{\text{corr}}(\phi) = \tilde{M}^{\mathcal{C}}\left(\,|\Phi_\phi\left(\mathcal{I}_s, \mathcal{I}_t\right) - \tilde{\mathcal{C}}| + \epsilon\right)^q. \tag{14}$$

Operator $|\cdot|$ denotes the $\ell_1$ norm, $q < 1$ (we set it to $q = 0.4$) and $\epsilon$ is a small constant. Ground-truth correspondences are denoted by $\tilde{\mathcal{C}}$. Since valid ground truth for all pixels is not available, we employ a ground-truth mask $\tilde{M}^{\mathcal{C}}$ to avoid propagating gradients through invalid pixels.

**Graph loss.** We impose an $l_2$-loss on node translations $\mathbf{t}$ (ground-truth rotations are not available):

$$\mathcal{L}_{\text{graph}}(\phi,\psi) = \tilde{M}^{\mathcal{V}}\left\|\left[\underbrace{\Omega\big(\Phi_\phi\left(\mathcal{I}_s, \mathcal{I}_t\right), \Psi_\psi\left(\mathcal{Z}_s, \mathcal{Z}'_t, \mathcal{H}\right), \mathcal{V}\big)}_{\mathcal{T}}\right]_{\mathbf{t}} - \tilde{\mathbf{t}}\right\|_2^2, \tag{15}$$

where $[\,\cdot\,]_{\mathbf{t}} : \mathbb{R}^{N\times 6} \rightarrow \mathbb{R}^{N\times 3}, \quad \mathcal{T} \mapsto [\mathcal{T}]_{\mathbf{t}} = \mathbf{t}$ extracts the translation part from the graph motion $\mathcal{T}$. Node translation ground-truth is denoted by $\tilde{\mathbf{t}}$ and $\tilde{M}^{\mathcal{V}}$ masks out invalid nodes. Please see the supplement for further details on how $\tilde{M}^{\mathcal{V}}$ is computed.

**Warp loss.** We have found that it is beneficial to use the estimated graph deformation $\mathcal{T}$ to deform the dense source point cloud $\mathcal{P}_s$ and enforce the result to be close to the source point cloud when deformed with the ground-truth scene flow $\tilde{\mathcal{S}}$:

$$\mathcal{L}_{\text{warp}}(\phi,\psi) = \tilde{M}^{\mathcal{S}}\left\|\mathrm{Q}\Big(\mathcal{P}_s, \underbrace{\Omega\big(\Phi_\phi\left(\mathcal{I}_s, \mathcal{I}_t\right), \Psi_\psi\left(\mathcal{Z}_s, \mathcal{Z}'_t, \mathcal{H}\right), \mathcal{V}\big)}_{\mathcal{T}}\Big) - (\mathcal{P}_s + \tilde{\mathcal{S}})\right\|_2^2. \tag{16}$$

Here, we extend the warping operation $\mathrm{Q}$ (Eq. 2) to operate on the dense point cloud $\mathcal{P}_s$ element-wise, and define $\tilde{M}^{\mathcal{S}}$ to mask out invalid points.

Note that We found it to be a more general notation to disentangle them (e.g., for scenarios where graph nodes are not sampled on the RGB-D frame).

### 4.5 Neural Non-rigid Tracking for 3D Reconstruction

We introduce our differentiable tracking module into the non-rigid reconstruction framework of Newcombe et al. [23]. In addition to the dense depth ICP correspondences employed in the original method, which help towards local deformation refinement, we employ a keyframe-based tracking objective. Without loss of generality, every 50th frame of the sequence is chosen as a keyframe, to

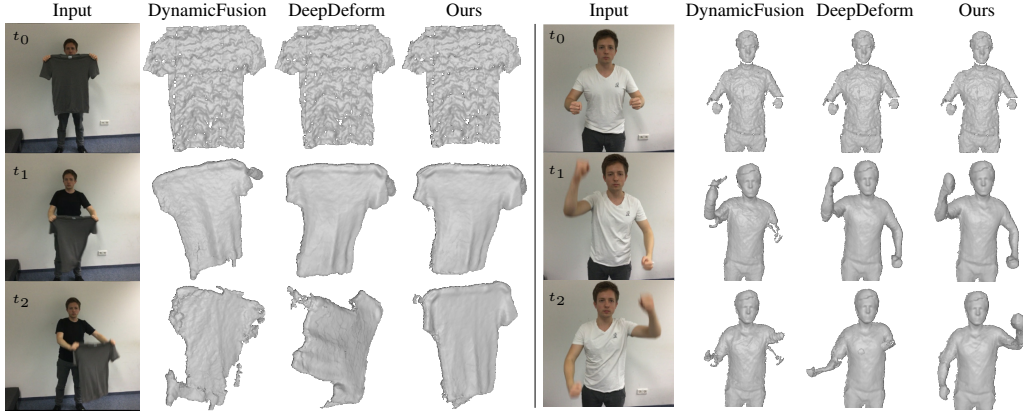

Figure 3: Qualitative comparison of our method with DynamicFusion [23] and DeepDeform [4] on test sequences from [4]. The rows show different time steps of the sequence.

which we establish dense correspondences including the respective weights. We apply a conservative filtering of the predicted correspondences based on the predicted correspondence weights using a fixed threshold $\delta = 0.35$ and re-weight the correspondences based on bi-directional consistency, i.e., keyframe-to-frame and frame-to-keyframe. Using the correspondence predictions and correspondence weights of valid keyframes ($> 50\%$ valid correspondences), the non-rigid tracking optimization problem is solved. The resulting deformation field is used to integrate the depth frame into the canonical volume of the object. We refer to the original reconstruction paper [23] for details regarding the fusion process.

## 5 Experiments

In the following, we evaluate our method quantitatively and qualitatively on both non-rigid tracking and non-rigid reconstruction. To this end, we use the DeepDeform dataset [4] for training, with the given 340-30-30 train-val-test split of RGB-D sequences. Both non-rigid tracking and reconstruction are evaluated on the hidden test set of the DeepDeform benchmark.

### 5.1 Training Scheme

The non-rigid tracking module has been implemented using the PyTorch library [24] and trained using stochastic gradient descent with momentum $0.9$ and learning rate $10^{-5}$. We use an Intel Xeon 6240 Processor and an Nvidia RTX 2080Ti GPU. The parameters of the dense correspondence prediction network $\phi$ are initialized with a PWC-Net model pre-trained on FlyingChairs [6] and FlyingThings3D [22]. We use a 10-factor learning rate decay every 10k iterations, requiring

Table 1: We evaluate non-rigid tracking on the DeepDeform dataset [4], showing the benefit of end-to-end differentiable optimizer losses and self-supervised correspondence weighting. We denote correspondence prediction as $\Phi_c$, $\Phi_{c+g}$ and $\Phi_{c+g+w}$, depending on which losses $\mathcal{L}_{corr}$, $\mathcal{L}_{graph}$, $\mathcal{L}_{warp}$ are used, and correspondence weighting as $\Psi_{supervised}$ and $\Psi_{self-supervised}$, either using an additional supervised loss or not.

| Model | EPE 3D (mm) | Graph Error 3D (mm) |
|---|---|---|
| $\Phi_c$ | 44.05 | 67.25 |
| $\Phi_{c+g}$ | 39.12 | 57.34 |
| $\Phi_{c+g+w}$ | 36.96 | 54.24 |
| $\Phi_c + \Psi_{supervised}$ | 28.95 | 36.77 |
| $\Phi_{c+g+w} + \Psi_{supervised}$ | 27.42 | 34.68 |
| $\Phi_{c+g+w} + \Psi_{self-supervised}$ | **26.29** | **31.00** |

about 30k iterations in total for convergence, with a batch size of $4$. For optimal performance, we first optimize the correspondence predictor $\Phi_\phi$ with $(\lambda_{\text{corr}}, \lambda_{\text{graph}}, \lambda_{\text{warp}}) = (5, 5, 5)$, without the weighting function $\Psi_\psi$. Afterwards, we optimize the weighting function parameters $\psi$ with $(\lambda_{\text{corr}}, \lambda_{\text{graph}}, \lambda_{\text{warp}}) = (0, 1000, 1000)$, while keeping $\phi$ fixed. Finally, we fine-tune both $\phi$ and $\psi$ together, with $(\lambda_{\text{corr}}, \lambda_{\text{graph}}, \lambda_{\text{warp}}) = (5, 5, 5)$.

## 5.2 Non-rigid Tracking Evaluation

For any frame pair $\mathcal{X}_{s,t}$ in the DeepDeform data [4], we define a deformation graph $\mathcal{G}$ by uniformly sampling graph nodes $\mathcal{V}$ over the source object in the RGB-D frame, given a segmentation mask of the former. Graph node connectivity $\mathcal{E}$ is computed using geodesic distances on a triangular mesh defined over the source depth map. As a pre-processing step, we filter out any frame pairs where more than $30\%$ of the source object is occluded in the target frame. In Table 1 non-rigid tracking performance is evaluated by the mean translation error over node translations $\mathbf{t}$ (Graph Error 3D), where the latter are compared to ground-truth with an $l_2$ metric. In addition, we evaluate the dense end-point-error (EPE 3D) between the source point cloud deformed with the estimated graph motion, $\mathrm{Q}(\mathcal{P}_s, \mathcal{T})$, and the source point cloud deformed with the ground-truth scene flow, $\mathcal{P}_s + \tilde{\mathcal{S}}$. To support reproducibility, we report the mean error metrics of multiple experiments, running every setting 3 times. We visualize the standard deviation with an error plot in the supplement.

We show that using graph and warp losses, $\mathcal{L}_{\text{graph}}$ and $\mathcal{L}_{\text{warp}}$, and differentiating through the non-rigid optimizer considerably improves both EPE 3D and Graph Error 3D compared to only using the correspondence loss $\mathcal{L}_{\text{corr}}$. Adding self-supervised correspondence weighting further decreases the errors by a large margin. Supervised outlier rejection with binary cross-entropy loss does bring an improvement compared to models that do not optimize for the weighting function $\Psi_\psi$ (please see supplemental material for details on this supervised training of $\Psi_\psi$). However, optimizing $\Psi_\psi$ in a *self-supervised* manner clearly outperforms the former supervised setup. This is due to the fact that, in the self-supervised scenario, gradients that flow from $\mathcal{L}_{\text{graph}}$ and $\mathcal{L}_{\text{warp}}$ through the differentiable solver $\Omega$ can better inform the optimization of $\Psi_\psi$ by minimizing the end-to-end alignment losses.

## 5.3 Non-rigid Reconstruction Evaluation

We evaluate the performance of our non-rigid reconstruction approach on the DeepDeform benchmark [4] (see Table 2). The evaluation metrics measure *deformation error*, a 3D end-point-error between tracked and annotated correspondences, and *geometry error*, which compares reconstructed shapes with annotated foreground object masks. Our approach performs about $8.9\%$ better than the state-of-the-art non-rigid reconstruction approach of Božič et al. [4] on the deformation metric. While our approach consistently shows better performance on both metrics, we also significantly lower the per-frame runtime to $27\,\text{ms}$ per keyframe, in contrast to [4], which requires $2299\,\text{ms}$. Thus, our approach can also be used with multiple keyframes at interactive frames rates, e.g., $90\,\text{ms}$ for 5 keyframes and $199\,\text{ms}$ for 10 keyframes.

Table 2: Our method achieves state-of-the-art non-rigid reconstruction results on the DeepDeform benchmark [4]. Both our end-to-end differentiable optimizer and the self-supervised correspondence weighting are necessary for optimal performance. Not only does our approach achieve lower deformation and geometry error compared to state of the art, our correspondence prediction is about $85\times$ faster.

| Method | Deformation error (mm) | Geometry error (mm) |
|---|---|---|
| DynamicFusion [23] | 61.79 | 10.78 |
| VolumeDeform [13] | 208.41 | 74.85 |
| DeepDeform [4] | 31.52 | 4.16 |
| Ours ($\Phi_c$) | 54.85 | 5.92 |
| Ours ($\Phi_{c+g+w}$) | 53.27 | 5.84 |
| Ours ($\Phi_c + \Psi_{\text{supervised}}$) | 40.21 | 5.39 |
| Ours ($\Phi_{c+g+w} + \Psi_{\text{self-supervised}}$) | **28.72** | **4.03** |

To show the influence of the different learned components of our method, we perform an ablation study by disabling either of our two main components: the end-to-end differentiable optimizer or the self-supervised correspondence weighting. As can be seen, our end-to-end trained method with self-supervised correspondence weighting demonstrates the best performance. Qualitatively, we show this in Figure 3. In contrast to DynamicFusion [23] and DeepDeform [4], our method is notably more robust in fast motion scenarios. Additional qualitative results and comparisons to the methods of Guo et al. [10] and Slavcheva et al. [26] are shown in the supplemental material.

## 6 Conclusion

We propose Neural Non-Rigid Tracking, a differentiable non-rigid tracking approach that allows learning the correspondence prediction and weighting of traditional tracking pipelines in an end-to-end manner. The differentiable formulation of the entire tracking pipeline enables back-propagation to the learnable components, guided by a loss on the tracking performance. This not only achieves notably improved tracking error in comparison to state-of-the-art tracking approaches, but also leads to better reconstructions, when integrated into a reconstruction framework like DynamicFusion [23]. We hope that this work inspires further research in the direction of neural non-rigid tracking and believe that it is a stepping stone towards fully differentiable non-rigid reconstruction.

## Broader Impact

Our paper presents learned non-rigid tracking. It is establishing the basis for the important research field of non-rigid tracking and reconstruction, which is needed for a variety of applications where man-machine and machine-environment interaction is required. These applications range from the field of augmented and virtual reality to autonomous driving and robot control. In the former, a precise understanding of dynamic and deformable objects is of major importance in order to provide an immersive experience to the user. Applications such as holographic calls would greatly benefit from research like ours. This, in turn, could provide society with the next generation of 3D communication tools. On the other hand, as a low-level building block, our work has no direct negative outcome, other than what could arise from the aforementioned applications.

## Acknowledgments and Disclosure of Funding

This work was supported by the ZD.B (Zentrum Digitalisierung.Bayern), the Max Planck Center for Visual Computing and Communications (MPC-VCC), a TUM-IAS Rudolf Mößbauer Fellowship, the ERC Starting Grant Scan2CAD (804724), and the German Research Foundation (DFG) Grant Making Machine Learning on Static and Dynamic 3D Data Practical.

## Footnotes

*Denotes equal contribution.

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
