[Supplementary Material]

# Neural Non-rigid Tracking
# –Supplementary Material–

**Aljaž Božič**[1]* 
aljaz.bozic@tum.de

**Pablo Palafox**[1]* 
pablo.palafox@tum.de

**Michael Zollhöfer**[2]    **Angela Dai**[1]    **Justus Thies**[1]    **Matthias Nießner**[1]

[1]Technical University of Munich        [2]Stanford University

## Contents

## A   Non-rigid Deformation Model

To represent the dense motion from a source to a target RGB-D frame, we adapt the embedded deformation model of Sumner et al. [8]. We uniformly sample graph nodes $\mathcal{V}$ over the source RGB-D frame (see Fig. 1), ensuring $\sigma$-coverage of the foreground object, i.e., the distance of every foreground point to the nearest graph node is at most $\sigma > 0$ (we set $\sigma = 0.05$ m).

Figure 1: Given a an object in the source RGB-D frame, we define a deformation graph $\mathcal{G}$ over the former. Nodes $\mathcal{V}$ (red spheres) are uniformly subsampled over the source RGB-D frame. Edges $\mathcal{E}$ (green lines) are computed between nodes based on geodesic connectivity among the latter.

For every node $v_i \in \mathcal{V}$, we estimate its global translation vector $\mathbf{t}_{\mathbf{v}_i} \in \mathbb{R}^3$ and local rotation matrix $\mathbf{R}_{\mathbf{v}_i} \in \mathbb{R}^{3 \times 3}$, represented in axis-angle notation as $\boldsymbol{\omega}_{\mathbf{v}_i} \in \mathbb{R}^3$. Using the deformation parameters $\mathcal{T} = (\boldsymbol{\omega}_{\mathbf{v}_1}, \mathbf{t}_{\mathbf{v}_1}, \ldots, \boldsymbol{\omega}_{\mathbf{v}_N}, \mathbf{t}_{\mathbf{v}_N})$ a 3D point $\mathbf{p} \in \mathbb{R}^3$ is deformed by interpolating the nodes' motion:

$$Q(\mathbf{p}, \mathcal{T}) = \sum_{\mathbf{v}_i \in \mathcal{V}} \alpha_{\mathbf{v}_i}^{\mathbf{P}} (e^{\widehat{\boldsymbol{\omega}}_{\mathbf{v}_i}}(\mathbf{p} - \mathbf{v}_i) + \mathbf{v}_i + \mathbf{t}_{\mathbf{v}_i}). \tag{1}$$

The weights $\alpha_{\mathbf{v}_i}^{\mathbf{P}} \in \mathbb{R}$ are called *skinning* weights and measure the influence of each node on the current point $\mathbf{p}$. They are computed as in DoubleFusion [11]:

$$\alpha_{\mathbf{v}_i}^{\mathbf{P}} = C e^{\frac{1}{2\sigma^2} ||\mathbf{v}_i - \mathbf{p}||_2^2}.$$

Here, $C$ denotes the normalization constant, ensuring that skinning weights add up to one for point $\mathbf{p}$:

$$\sum_{\mathbf{v}_i \in \mathcal{V}} \alpha_{\mathbf{v}_i}^{\mathbf{P}} = 1.$$

For each node $\mathbf{v}_i \in \mathcal{V}$, we represent its rotation in axis-angle notation as $\boldsymbol{\omega}_{\mathbf{v}_i} \in \mathbb{R}^3$. This representation has singularities for larger angles, i.e., two different vectors $\boldsymbol{\omega}$ and $\boldsymbol{\omega}'$ can represent the same rotation (for example keeping the same axis and increasing the angle by $2\pi$ results in identical rotation). To avoid singularities, we decompose the rotation matrix into $e^{\widehat{\boldsymbol{\omega}}_{\mathbf{v}_i}} = e^{\widehat{\boldsymbol{\epsilon}}_{\mathbf{v}_i}} \mathbf{R}_{\mathbf{v}_i}$ with $\boldsymbol{\epsilon}_{\mathbf{v}_i} = 0$, therefore optimizing only for delta rotations that have rather small rotation angles.

## B   Differentiable Non-rigid Optimization

Our non-rigid optimization is based on the Gauss-Newton algorithm and minimizes an energy formulation that is based on three types of residual components: the 2D reprojection term, the depth term and the regularization term of the non-rigid deformation.

For a pixel $\mathbf{u} \in \Pi_s \subset \mathbb{R}^2$ and graph edge $(\mathbf{v}_i, \mathbf{v}_j) \in \mathcal{E}$, we define such terms as:

$$r_{2D}^{\mathbf{u}}(\mathcal{T}) = w_{\mathbf{u}}\big(\pi_{\mathbf{c}}(Q(\mathbf{p}_{\mathbf{u}}, \mathcal{T})) - \mathbf{c}_{\mathbf{u}}\big)$$
$$r_{depth}^{\mathbf{u}}(\mathcal{T}) = w_{\mathbf{u}}\big([Q(\mathbf{p}_{\mathbf{u}}, \mathcal{T})]_z - [P_t(\mathbf{c}_{\mathbf{u}})]_z\big)$$
$$r_{reg}^{\mathbf{v}_i, \mathbf{v}_j}(\mathcal{T}) = e^{\widehat{\boldsymbol{\omega}}_{\mathbf{v}_i}}(\mathbf{v}_j - \mathbf{v}_i) + \mathbf{v}_i + \mathbf{t}_{\mathbf{v}_i} - (\mathbf{v}_j + \mathbf{t}_{\mathbf{v}_j}),$$

where $\mathbf{c_u} \in \mathbb{R}^2$ and $w_\mathbf{u} \in \mathbb{R}$ represent the predicted correspondence and the importance weight, respectively; and $\mathbf{p_u} = \pi_\mathbf{c}^{-1}(\mathbf{u}, d_\mathbf{u})$ is a 3D point corresponding to the pixel $\mathbf{u}$ with depth value $d_\mathbf{u}$.

The Gauss-Newton method is an iterative scheme. In every iteration $n$, we compute the Jacobian matrix $\mathbf{J}_n$ and the residual vector $\mathbf{r}_n$, and get a solution increment $\Delta\mathcal{T}$ by solving the normal equations:

$$\mathbf{J}_n^T\mathbf{J}_n\Delta\mathcal{T} = -\mathbf{J}_n^T\mathbf{r}_n.$$

The construction of the Jacobian matrix $\mathbf{J} \in \mathbb{R}^{(3|\mathcal{C}|+3|\mathcal{E}|)\times 6N}$, consisting of partial derivatives of the residual vector $\mathbf{r}^{3|\mathcal{C}|+3|\mathcal{E}|}$ with respect to deformation parameters $\mathcal{T} = (\boldsymbol{\omega}_{\mathbf{v}_1}, \mathbf{t}_{\mathbf{v}_1}, \ldots, \boldsymbol{\omega}_{\mathbf{v}_N}, \mathbf{t}_{\mathbf{v}_N}) \in \mathbb{R}^{6N}$ is detailed in Section B.1. The linear system is solved using LU decomposition. To enable differentiation through the entire Gauss-Newton solver, we have to ensure that the linear solve is differentiable. We detail the differentiable linear solve operation in Section B.2.

## B.1 Partial Derivatives

In the following, we derive the partial derivatives of the residual vector $\mathbf{r}$ with respect to $\boldsymbol{\epsilon}_{\mathbf{v}_i}$ and $\mathbf{t}_{\mathbf{v}_i}$ of every node $\mathbf{v}_i$, to construct the Jacobian matrix $\mathbf{J}$.

To simplify notation, we define the rotation operator that takes as input an angular velocity vector $\boldsymbol{\epsilon} \in \mathbb{R}^3$, rotation matrix $\mathbf{R} \in \mathbb{R}^{3\times 3}$ and point $\mathbf{p} \in \mathbb{R}^3$ and outputs the rotated point:

$$\mathrm{R}(\boldsymbol{\epsilon}, \mathbf{R}, \mathbf{p}) = e^{\widehat{\boldsymbol{\epsilon}}}\mathbf{R}\mathbf{p}.$$

To compute the partial derivative with respect to $\boldsymbol{\epsilon}$, we follow the derivation from Blanco [2]:

$$\frac{\partial \mathrm{R}(\boldsymbol{\epsilon}, \mathbf{R}, \mathbf{p})}{\partial \boldsymbol{\epsilon}}\bigg|_{\boldsymbol{\epsilon}=0} = -\widehat{\mathbf{R}\mathbf{p}}$$

Here, the $\widehat{\cdot}$-operator creates a $3 \times 3$ skew-symmetric matrix from a 3-dimensional vector.

The rotation operator $\mathrm{R}(\boldsymbol{\epsilon}, \mathbf{R}, \mathbf{p})$ is a core part of the warping operator $\mathrm{Q}(\mathbf{p}, \mathcal{T})$. It follows that partial derivatives of a warping operator $\mathrm{Q}(\mathbf{p}, \mathcal{T})$ with respect to $\boldsymbol{\epsilon}_{\mathbf{v}_i}$ and $\mathbf{t}_{\mathbf{v}_i}$ for every node $\mathbf{v}_i$ can be computed as

$$\frac{\partial \mathrm{Q}(\mathbf{p}, \mathcal{T})}{\partial \boldsymbol{\epsilon}_{\mathbf{v}_i}} = -\alpha_{\mathbf{v}_i}^{\mathbf{p}}\widehat{\mathbf{R}_{\mathbf{v}_i}(\mathbf{p} - \mathbf{v}_i)},$$

$$\frac{\partial \mathrm{Q}(\mathbf{p}, \mathcal{T})}{\partial \mathbf{t}_{\mathbf{v}_i}} = \alpha_{\mathbf{v}_i}^{\mathbf{p}}\mathbf{I}.$$

Another building block of our optimization terms is the perspective projection $\pi_\mathbf{c}$ with intrinsic parameters $\mathbf{c} = (f_x, f_y, c_x, c_y)$:

$$\pi_\mathbf{c} : \mathbb{R}^3 \to \mathbb{R}^2,$$

$$\pi_\mathbf{c}\left(\begin{bmatrix} x \\ y \\ z \end{bmatrix}\right) = \begin{bmatrix} f_x\dfrac{x}{z} + c_x \\ f_y\dfrac{y}{z} + c_y \end{bmatrix},$$

whose partial derivatives with respect to the point $\mathbf{p} = (x, y, z)^{\mathrm{T}}$ are derived as

$$\frac{\partial \pi_\mathbf{c}(\mathbf{p})}{\partial \mathbf{p}} = \begin{bmatrix} \dfrac{f_x}{z} & 0 & -\dfrac{f_x x}{z^2} \\ 0 & \dfrac{f_y}{z} & -\dfrac{f_y y}{z^2} \end{bmatrix}.$$

By applying the chain rule, derivatives of all three optimization terms are computed.

**Derivative of 2D reprojection term.** For a pixel $\mathbf{u} \in \Pi_s \subset \mathbb{R}^2$ and its corresponding 3D point $\mathbf{p_u}$, we derive partial derivatives of $r_{2D}^{\mathbf{u}}(\mathcal{T})$ as follows:

$$\frac{\partial r_{2D}^{\mathbf{u}}(\mathcal{T})}{\partial \boldsymbol{\epsilon}_{\mathbf{v}_i}} = -w_{\mathbf{u}} \alpha_{\mathbf{v}_i}^{\mathbf{p_u}} \begin{bmatrix} \dfrac{f_x}{\mathbf{p_u^z}} & 0 & -\dfrac{f_x \mathbf{p_u^x}}{(\mathbf{p_u^z})^2} \\ 0 & \dfrac{f_y}{\mathbf{p_u^z}} & -\dfrac{f_y \mathbf{p_u^y}}{(\mathbf{p_u^z})^2} \end{bmatrix} \widehat{\mathbf{R}_{\mathbf{v_i}}(\mathbf{p_u} - \mathbf{v}_i)},$$

$$\frac{\partial r_{2D}^{\mathbf{u}}(\mathcal{T})}{\partial \mathbf{t}_{\mathbf{v}_i}} = w_{\mathbf{u}} \alpha_{\mathbf{v}_i}^{\mathbf{p_u}} \begin{bmatrix} \dfrac{f_x}{\mathbf{p_u^z}} & 0 & -\dfrac{f_x \mathbf{p_u^x}}{(\mathbf{p_u^z})^2} \\ 0 & \dfrac{f_y}{\mathbf{p_u^z}} & -\dfrac{f_y \mathbf{p_u^y}}{(\mathbf{p_u^z})^2} \end{bmatrix}.$$

**Derivative of depth term.** When computing the partial derivatives of the depth term $r_{\text{depth}}^{\mathbf{u}}(\mathcal{T})$, we need to additionally apply the projection to the $z$-component in the chain rule:

$$\frac{\partial r_{\text{depth}}^{\mathbf{u}}(\mathcal{T})}{\partial \boldsymbol{\epsilon}_{\mathbf{v}_i}} = -w_{\mathbf{u}} \alpha_{\mathbf{v}_i}^{\mathbf{p_u}} \begin{bmatrix} 0 & 0 & 1 \end{bmatrix} \widehat{\mathbf{R}_{\mathbf{v_i}}(\mathbf{p_u} - \mathbf{v}_i)},$$

$$\frac{\partial r_{\text{depth}}^{\mathbf{u}}(\mathcal{T})}{\partial \mathbf{t}_{\mathbf{v}_i}} = w_{\mathbf{u}} \alpha_{\mathbf{v}_i}^{\mathbf{p_u}} \begin{bmatrix} 0 & 0 & 1 \end{bmatrix}.$$

**Derivative of regularization term.** For a graph edge $(\mathbf{v}_i, \mathbf{v}_j) \in \mathcal{E}$, the partial derivatives of $r_{\text{reg}}^{\mathbf{v}_i, \mathbf{v}_j}(\mathcal{T})$ with respect to $\boldsymbol{\epsilon}_{\mathbf{v}_i}, \mathbf{t}_{\mathbf{v}_i}, \boldsymbol{\epsilon}_{\mathbf{v}_j}, \mathbf{t}_{\mathbf{v}_j}$ are computed as:

$$\frac{\partial r_{\text{reg}}^{\mathbf{v}_i, \mathbf{v}_j}(\mathcal{T})}{\partial \boldsymbol{\epsilon}_{\mathbf{v}_i}} = -\widehat{\mathbf{R}_{\mathbf{v_i}}(\mathbf{v}_j - \mathbf{v}_i)}, \qquad\qquad \frac{\partial r_{\text{reg}}^{\mathbf{v}_i, \mathbf{v}_j}(\mathcal{T})}{\partial \boldsymbol{\epsilon}_{\mathbf{v}_j}} = \mathbf{0},$$

$$\frac{\partial r_{\text{reg}}^{\mathbf{v}_i, \mathbf{v}_j}(\mathcal{T})}{\partial \mathbf{t}_{\mathbf{v}_i}} = \mathbf{I}, \qquad\qquad \frac{\partial r_{\text{reg}}^{\mathbf{v}_i, \mathbf{v}_j}(\mathcal{T})}{\partial \mathbf{t}_{\mathbf{v}_j}} = -\mathbf{I}.$$

## B.2 Differentiable Linear Solve Operation

To simplify the notation, in the following we use $\mathbf{A} = \mathbf{J}_n^T \mathbf{J}_n$, $\mathbf{b} = -\mathbf{J}_n^T \mathbf{r}_n$ and $\mathbf{x} = \Delta \mathcal{T}$, which results in the linear system of the form

$$\mathbf{A}\mathbf{x} = \mathbf{b}. \tag{2}$$

For matrix $\mathbf{A} \in \mathbb{R}^{6N \times 6N}$ and vectors $\mathbf{b} \in \mathbb{R}^{6N}$ and $\mathbf{x} \in \mathbb{R}^{6N}$ we define the linear solve operation as

$$\Lambda : \mathbb{R}^{6N \times 6N} \times \mathbb{R}^{6N} \to \mathbb{R}^{6N}, \quad (\mathbf{A}, \mathbf{b}) \mapsto \mathbf{A}^{-1}\mathbf{b} = \mathbf{x}. \tag{3}$$

In order to compute the derivative of the linear solve operation, we follow the analytic derivative formulation of Barron and Poole [1]. If we denote the partial derivative of the loss $\mathcal{L}$ with respect to linear system solution $\mathbf{x}$ as $\frac{\partial \mathcal{L}}{\partial \mathbf{x}}$, we can compute the partial derivatives with respect to matrix $\mathbf{A}$ and vector $\mathbf{b}$ as:

$$\frac{\partial \mathcal{L}}{\partial \mathbf{b}} = \mathbf{A}^{-1} \frac{\partial \mathcal{L}}{\partial \mathbf{x}}, \qquad \frac{\partial \mathcal{L}}{\partial \mathbf{A}} = \left( \mathbf{A}^{-1} \frac{\partial \mathcal{L}}{\partial \mathbf{x}} \right) \mathbf{x}^{\mathrm{T}} = -\frac{\partial \mathcal{L}}{\partial \mathbf{b}} \mathbf{x}^{\mathrm{T}}. \tag{4}$$

Thus, the computation of $\frac{\partial \mathcal{L}}{\partial \mathbf{b}}$ requires solving a linear system with matrix $A$. To solve this system, we re-use the LU decomposition from the forward pass.

## B.3 Ablations

We experimented with different design choices for our solver.

**ARAP edge re-weighting.** In non-rigid tracking, it is possible to weight ARAP terms for every graph edge differently, depending on the distance between the nodes. In our method, we sample nodes uniformly on the surface, thus, all edges have similar length ($7.13 \pm 1.38$ cm). Hence, edge re-weighting changes EPE 3D only marginally: $0.8\%$ lower EPE 3D and $1.7\%$ lower Graph Error 3D.

**Nearest-neighbor vs. bilinear depth sampling.**   When querying depth after predicting 2D correspondences, we found bilinear sampling to perform better, with $5.8\%$ lower EPE 3D and $6.29\%$ lower Graph Error 3D compared to nearest-neighbor sampling.

**Influence of graph density.**   We sample graph nodes with $5\,\mathrm{cm}$ node coverage, which fits well with our setup of $11\,\mathrm{GiB}$ for training. Using coarser graphs, with $10\,\mathrm{cm}$ and $15\,\mathrm{cm}$ node coverage resulted in poorer performance: $5.49\%$ and $8.33\%$ higher EPE 3D, as well as $7.34\%$ and $28.46\%$ higher Graph Error 3D, respectively. In turn, the memory footprint on the GPU during training (with batch size 4) decreases with node coverage: $10\,513\,\mathrm{MiB}$, $6153\,\mathrm{MiB}$ and $5931\,\mathrm{MiB}$ for $5\,\mathrm{cm}$, $10\,\mathrm{cm}$ and $15\,\mathrm{cm}$ node coverage, respectively.

**Number of optimization steps.**   We empirically found 3 solver iterations to be the best compromise between performance and computational cost. Unrolling 3 solver iterations instead of 1 / 2, results in $24.9\%$ / $0.16\%$ lower EPE 3D and $22.7\%$ / $0.23\%$ lower Graph Error 3D. Using 4 iterations only improves slightly with respect to 3 ($0.04\%$ lower EPE 3D, $0.03\%$ lower Graph Error 3D). More than 4 does not change performance notably.

**Warp loss as a superset of graph loss.**   Note that since we sample graph nodes on depth maps, the graph loss (Eq. 15 in the paper) is in practice a subset of the warp loss (Eq. 16 in the paper). However, we found it to be a more general notation to disentangle them. For instance, this notation is helpful for scenarios where graph nodes are not sampled on the RGB-D frame.

## C    Losses

In this section, we provide implementation details related to the training losses used for end-to-end optimization. Following the architecture of Sun et al. [9], our correspondence prediction function $\Phi$ computes a hierarchy of correspondence predictions, instead of only the correspondences at the highest resolution. These predictions are used to compute the correspondence loss in a coarse-to-fine fashion (see Section C.1). To achieve numerically stable optimization (Gauss-Newton solver) during training, a ground-truth mask is needed for the graph loss. In Section C.2, we detail how to ensure stable optimization by filtering invalid graph nodes.

In the ablation studies, we also included a comparison to a weight function $\Psi$ that is trained in a supervised manner (see Table 1 in the main paper). Section C.3 details the training of this baseline using a supervised binary cross-entropy loss.

### C.1    Coarse-to-fine Correspondence Loss

The design of our correspondence prediction function $\Phi$ follows the PWC-Net [9] architecture that predicts the correspondences in a coarse-to-fine manner. Initially the correspondences are predicted at a coarse resolution of $10 \times 7$ px, and then refined to a resolution of $20 \times 14$ px, etc. In total, there are $L = 5$ levels in the correspondence hierarchy, and the finest level predictions are used in the differentiable non-rigid optimization. The correspondence loss is applied on every level $l$, by bilinear downsampling of the groundtruth correspondences $\tilde{\mathcal{C}}$ to a coarser resolution, resulting in $\tilde{\mathcal{C}}^l$. Similarly, the ground-truth mask matrix $\tilde{M}^{\mathcal{C}}$ is downsampled to a coarser version $\tilde{M}^{\mathcal{C}^l}$, to avoid propagating gradients through invalid pixels. For each training sample $(\mathcal{I}_s, \mathcal{I}_t)$ and every level $l$, we therefore compute ground-truth correspondences $\tilde{\mathcal{C}}^l$ and the ground-truth mask matrix $\tilde{M}^{\mathcal{C}^l}$. At every level $l$ the correspondence loss has the following form:

$$\mathcal{L}^l_{\mathrm{corr}}(\phi) = \tilde{M}^{\mathcal{C}^l} (\, |\Phi^l_\phi (\mathcal{I}_s, \mathcal{I}_t) - \tilde{\mathcal{C}}^l| + \epsilon)^q. \tag{5}$$

With $q < 1$ (in our case $q = 0.4$) and $\epsilon$ being a small constant.

### C.2    Numerically Stable Optimization

The non-rigid tracking optimization objective includes data (correspondence and depth) terms and a regularization (ARAP) term. If we only use regularization term, the optimization problem becomes ill-posed, since any rigid transformation of all graph nodes has no effect on the regularization term.

In order to satisfy memory limits, we do not use all pixel correspondences $\mathcal{C}$ at training time, but instead randomly sample 10k correspondences.

The edge set $\mathcal{E}$ of the deformation graph $\mathcal{G} = (\mathcal{V}, \mathcal{E})$ is computed by connecting each graph node with $K = 8$ nearest nodes, using geodesic distances on the depth map mesh as a metric. This can lead to multiple disconnected graph components, i.e., different node clusters. To ensure the optimization problem is well-defined, we ensure that we have a minimum number of correspondences in each node cluster. In our experiments we filter out all node clusters with less than 2000 correspondences. This filtering has to be reflected in the loss computation. Thus, we define the mask matrix $\tilde{M}^{\mathcal{V}}$ to have zeros for nodes from filtered clusters, which prevents gradient back-propagation through invalidated graph nodes.

### C.3 Supervised Weight Network Baseline

As a baseline, we introduce a model where we supervise the optimization of the weighting function $\Psi$. The ground-truth correspondence weighting $\tilde{\mathcal{W}} \in \mathbb{R}^{H \times W \times 1}$ for this supervision is generated by comparing current correspondence predictions $\mathcal{C}$ against the ground-truth correspondences $\tilde{\mathcal{C}}$. We compare 3D distances between correspondences, using the target depth map $\mathcal{D}_t$ to query corresponding depth values. A pixel in the ground-truth weighting $\tilde{\mathcal{W}}$ is assigned a 1 or 0 depending on the correspondence error. Optimal performance was achieved by assigning 1 to correspondences that are at most $0.1\,\mathrm{m}$ away from groundtruth, and 0 to correspondences that are at least $0.3\,\mathrm{m}$ away from groundtruth, without propagating any gradient through remaining correspondence weights. Binary cross-entropy loss is used to optimize $\Psi$ in this supervised setting.

## D Reproducibility

### D.1 Time and Memory Complexity

**Correspondence Prediction $\Phi$.** Function $\Phi$ scales as a standard convolutional neural network linearly with the number of pixels in the input image (both in time and memory complexity). The number of layers and kernel sizes are independent of the input an, thus, constant. Our correspondence prediction network $\Phi$ consists of 55 layers with a total number of 9.374M parameters. The processing of an image of resolution $640 \times 480$ takes 21.6 ms.

**Correspondence Weighting $\Psi$.** Function $\Psi$ is a convolutional neural network with a fixed number of layers and kernels, and, thus, has a complexity that is linear in the number of pixels of the input image. In total the network consists of 316K learnable parameters that are distributed among 7 layers. For a forward pass with an image of resolution $640 \times 480$ the network takes 5.5 ms.

**Differentiable Optimizer.** Time and memory complexity of our differentiable Gauss-Newton solver is dominated by two operations: matrix-matrix multiplication of $\mathbf{J}^{\mathrm{T}}$ and $\mathbf{J}$ and LU decomposition of $\mathbf{J}^{\mathrm{T}}\mathbf{J}$ matrix. For a matrix $\mathbf{J} \in \mathbb{R}^{(3|\mathcal{C}|+3|\mathcal{E}|) \times 6N}$ the matrix-matrix multiplication $\mathbf{J}^{\mathrm{T}}\mathbf{J}$ has a time complexity of $\mathrm{O}(N^2 \cdot (|\mathcal{C}| + |\mathcal{E}|))$ and memory complexity of $\mathrm{O}(N \cdot (|\mathcal{C}| + |\mathcal{E}|))$. We denoted the number of correspondences with $|\mathcal{C}|$, the number of graph edges with $|\mathcal{E}|$ and the number of graph nodes with $N$. On the other hand, the time and memory complexity of LU decomposition of a matrix $\mathbf{J}^{\mathrm{T}}\mathbf{J} \in \mathbb{R}^{6N \times 6N}$ is $\mathrm{O}(N^3)$ and $\mathrm{O}(N^2)$, respectively. Note that LU is dominated by matrix-matrix multiplication; in theory there exist algorithms better then $n^3$, like $n^{2.376}$ based on the Coppersmith–Winograd algorithm [4]. The total time complexity is, therefore, $\mathrm{O}(N^2 \cdot (|\mathcal{C}| + |\mathcal{E}|) + N^3)$ and memory complexity is $\mathrm{O}(N \cdot (|\mathcal{C}| + |\mathcal{E}|) + N^2)$.

### D.2 Training Details

For reproducibility, the analysis of the achieved performance of the network with different training runs is important. In the main paper (Table 1), we show an ablation study of our method and report average test errors of 3 training runs. In Figure 2 the corresponding standard deviations are plotted. As can be seen, the training of our network is stable and results in small variations in performance. For all experiments we used an Intel Xeon 6240 Processor with 18 cores and an Nvidia GeForce RTX 2080Ti GPU. A typical power consumption of Nvidia 2080Ti GPU is around 280 Watts. Network

Figure 2: Plots of non-rigid tracking EPE 3D (left) and Graph Error 3D (right) values, together with standard deviation bars. Corresponds to Table 1 in the main paper.

experiments were run for 30k iterations with batch size $4$, requiring in total about $14$ hours till convergence.

### D.3 Keyframe-based Non-rigid Reconstruction

To achieve robust non-rigid reconstruction, we propose the usage of a keyframe-based strategy. We explored different keyframe sampling, as shown in Table 1, and opted for sampling a keyframe every 50 frames in our setup. For every keyframe, the correspondences to the latest frame in the video are predicted. Our neural non-rigid tracker provides us with correspondence $\mathbf{c_u}$ and an importance weight $w_{\mathbf{u}} \in [0, 1]$ for every pixel $\mathbf{u} \in \Pi_s \subset \mathbb{R}^2$. We invalidate all correspondences of a keyframe with $w_{\mathbf{u}} < \delta$ (we set $\delta = 0.35$ in all experiments). In case of large occlusions between the current frame and the keyframe, many correspondences are invalid. If $50\%$ of the correspondences are invalid, we completely ignore the keyframe, which leads to less outliers and faster runtime.

Table 1: We evaluate how the deformation error (mm) varies with the keyframe sampling. More frames means lower keyframe sampling rate, i.e., larger frame-to-frame motion.

| Keyframe density | Deformation error (mm) |
| --- | --- |
| DeepDeform [3] (filtering w/ neighboring frames) | 31.52 |
| Ours: keyframe every 100 frames | 30.70 |
| Ours: keyframe every 75 frames | 29.68 |
| Ours: keyframe every 50 frames | **28.72** |

In addition, we apply correspondence reweighting based on cycle consistencies. Specifically, we enforce bi-directional consistency and multi-keyframe consistency. *Bi-directional consistency* enables us to detect self-occlusions between a keyframe and current frame. Correspondences are predicted in both directions keyframe-to-frame and frame-to-keyframe. If following the correspondence in forward keyframe-to-frame and afterwards in backward frame-to-keyframe direction results in a 3D error larger than $0.20\,\mathrm{m}$, we reject the correspondence. For *multi-keyframe consistency*, multiple keyframe-to-frame predictions are estimated that correspond to the same 3D point in the canonical volume and the mean prediction value is computed. If any of the predictions is more than $0.15\,\mathrm{m}$ away from the mean value, we reject all correspondences for a given 3D canonical point.

## E  Benchmark Results

In Figure 3 we provide a screenshot of the currently best-performing, non-rigid reconstruction methods on the DeepDeform [3] benchmark. Please visit `http://kaldir.vc.in.tum.de/deepdeform_benchmark/benchmark_reconstruction`.

## Non-rigid Reconstruction Benchmark

This table lists the benchmark results for the Non-rigid Reconstruction scenario.

| Method | Info | Geometry error (cm) ▽ | Deformation error (cm) ▼ |
|---|---|---|---|
| Neural Non-rigid Tracking | | **0.403** | **2.872** |
| DeepDeform | | 0.416 | 3.152 |
| Aljaž Božič, Michael Zollhöfer, Christian Theobalt, Matthias Nießner: DeepDeform: Learning Non-rigid RGB-D Reconstruction with Semi-supervised Data. CVPR 2020 | | | |
| DynamicFusion | | 1.078 | 6.179 |
| Richard Newcombe, Dieter Fox, Steve Seitz: DynamicFusion: Reconstruction and Tracking of Non-rigid Scenes in Real-Time. CVPR 2015 | | | |
| VolumeDeform | | 7.485 | 20.841 |
| Matthias Innmann, Michael Zollhöfer, Matthias Nießner, Christian Theobalt, Marc Stamminger: VolumeDeform: Real-time Volumetric Non-rigid Reconstruction. ECCV 2016 | | | |

Figure 3: Screenshot of non-rigid reconstruction results on DeepDeform [3] benchmark (taken on 11th June 2020).

## F   Additional Results

In the following, we present additional qualitative results of our method in comparison to state-of-the-art methods. Figure 4 shows a comparison to Guo et al. [5]. As can be seen, our method better handles non-rigid movements with fast motions (t-shirt) and occlusions (arm).

In Figure 5, we show the results of applying our method on test sequences of the DeepDeform dataset [3], and compare to the results of Slavcheva et al. [7]. The reconstruction of our method leads to more complete and smooth meshes. Note that the results of both methods [5] and [7] were kindly provided by the authors.

We show qualitative reconstruction results of our method on VolumeDeform [6] sequences in Figure 6. Our method can robustly reconstruct these RGB-D sequences, despite the fact that a Kinect sensor was used to record them, whereas our training data was obtained using a Structure IO sensor. This shows that our network predictions can generalize to a different structured-light sensor input.

We also compared to DoubleFusion [11] and BodyFusion [10], which focus on human body reconstruction by assuming human body prior. We were able to compare on a sequence provided by [10], and even without assuming any explicit shape or motion priors, we achieve competitive performance. In particular, our method achieves an average tracking error of $0.0317\,\mathrm{m}$, while BodyFusion and DoubleFusion achieve $0.0227\,\mathrm{m}$ and $0.0221\,\mathrm{m}$, respectively. Note that these methods cannot reconstruct non-human sequences, unlike our approach.

Additionally, in Figure 7 we show texturing results, computed by aggregating color images over 100 frames of motion into a voxel grid.

Figure 4: Qualitative comparison of our method to MonoFVV [5] (test sequences from [3]).

Figure 5: Qualitative comparison of our method to KillingFusion [7] (test sequences from [3]).

Figure 6: Qualitative reconstruction results on VolumeDeform [6] sequences.

Figure 7: Texturing results, computed by aggregating color images over 100 frames of motion into a voxel grid.

## Footnotes

*Denotes equal contribution.