[Reviews · NeurIPS 2020]

Review 1

Summary and Contributions: The authors introduce a non-rigid tracker for given RGB-D frames. Combining correspondence networks with differentiable optimization solver enables the proposed network to be trained in an end-to-end fashion. Additional module that estimates correspondence weights allow robust tracking.

Strengths: 1. Integrating the ideas of state-of-the-art methods (correspondence prediction and optimization solver) in a differentiable way 2. The paper is generally well written and organized with promising results

Weaknesses: 1. Originality: Given that similar ideas have appeared in the previous works (correspondence prediction [4] and differentiable optimization solver [15]), the proposed method would be considered as a combination of them with some modifications. For example, dense correspondences are estimated rather than sparse ones in [4], but this is formulated simply leveraging previous state-of-the-art optical flow technique. The used energy terms and loss functions are similarly defined to [15] except that eq. (9) is based on reconstruction of "displacements" rather than "features" in [15]. These make me hard to find new technical insights from the paper. 2. Ablation study: Under severe non-rigid motions, dense correspondences would not be always beneficial than sparse ones due to their robustness to geometric variations. Since the experiments are conducted solely on benchmark of [4] which seems to have mild variations along time axis, additional evaluation and analysis will be expected under larger geometric variations (ex. by decreasing sampling rate of keyframe).

Correctness: It would be nice to highlight the differences between the proposed method and [4], [15].

Clarity: The paper is generally well-written and structured clearly.

Relation to Prior Work: See the Correctness section. Authors might want to cite these as related self-supervised RGB tracking works: - Learning Correspondence from the Cycle-consistency of Time (CVPR 2019) - Joint-task Self-supervised Learning for Temporal Correspondence (NeurIPS 2019) - MAST: A Memory-Augmented Self-supervised Tracker (CVPR 2020)

Reproducibility: Yes

Additional Feedback: Instead of learning correspondence confidences from reconstruction loss given ground-truth correspondences, It would be interesting by considering introspection to down weight ambiguous matches as proposed in [A]. [A] Supervising the New with the Old: Learning SFM from SFM (ECCV 2018) -- After rebuttal -- After reading the authors rebuttal and seeing the comments from the other reviewers, I still think It's hard to find new technical insights from this paper. Given that [15] learns non-rigid tracking in an end-to-end fashion through the differentiable solver, this paper combines [15] with non-rigid matches similar to [4]. Thus, as claimed in the rebuttal L5-12, the technical contributions lie on 1) estimating correspondence confidences and 2) using dense correspondence rather sparse one. Despite of the expected good results, their formulations are straight-forward without tailored modifications for RGB-D tracking as R4 also pointed out. Specifically, learning correspondence confidences by weighting the final matching objectives (as in Eq.(9) and (10) ) is a classical technique [a,b] to discount the effects of outlier matches. In general, learning intermediate attentions via a weighted loss for a proxy task [c,d,e] is a well-known strategy for boosting the performance. In terms of architecture, the proposed correspondence weighting networks just consist of sequential 7 convolutional layers (L132-135 in supp.). Additionally, the estimation of dense correspondences is done by simply adapting previous SOTA optical flow algorithms without any changes. [a] View Synthesis by Appearance Flow, ECCV 2016 [b] Learning Dense Correspondence via 3D-guided Cycle Consistency, CVPR 2016 [c] Spatial Transformer Networks, NeurIPS 2015 [d] Dynamic Filter Networks, NeurIPS 2016 [e] Deformable Convolutional Networks, ICCV 2017


Review 2

Summary and Contributions: The authors present a deep structured model for non-rigid tracking. The key idea is to combine deep learning with existing optimization methods and takes the best of both worlds. Since the optimization algorithm can be written as a neural network layer without learnable parameters, the model can be trained in an end-to-end fashion. After joint training, the network learns to re-weight the correspondences in a self-supervised fashion. The optimization solver can effectively estimate the non-rigid transform over the pre-defined graph. They achieve state-of-the-art performance while being 86 times faster.

Strengths: - The authors present a good pipeline of non-rigid tracking at the instance level. The energy optimization steps ensure the deformation graph to be locally rigid and are unrolled into end-to-end learning. Although every module is not entirely novel in its own, the experiments show that combining them can achieve state-of-the-art performance. (For instance, [18] learns to predict the damping factor for LM solver; [19] unrolls GN as recurrent neural network.) - Comparing to prior art [4], the authors significantly reduce the computational time of correspondence prediction by 85 times - Writing and experiments are good in general. The implementations are technically sound in details. Based on the writing, the method should be reproducible from the writing.

Weaknesses: - It seems to me that the warp loss is the super set of the graph loss. When p \in P_s is a node of the graph V, the warp loss reduces to the graph loss. Therefore the two loss functions can be viewed as one, with the weightings of each point cloud being different, depending on whether you are a graph node or not. - Do the authors re-weight the term for each edge in ARAP? Theoretically, one should assign different weightings to different edges unless the edges are all more or less of the same length. Can the authors comment more on this? - How does the graph size affect the method? How do the authors decide this? My guess is that increasing the number of nodes shall improve the performance, but it will also increase the computational cost. How do the authors balance this? - Seems like the method relies on a decent segmentation model to build the graph. What will happen if the estimated mask is noisy? To what extent will the proposed method be affected by it. - How does the number of optimization steps affect the performance? Will the performance degrade if you unroll more steps during inference? - I don't think bilinear sampling the depth image is a good idea. To my knowledge, using nearest neighbor is better. Otherwise it may result in some smearing effect at the boundaries. Would be great if the authors can comment on this. *** extra suggestions after the rebuttal *** - I agree with other reviewers that the authors should not claim "GN solver" as their contribution as a lot of work have done very similar things before (eg [18, 19]). Rather they should focus their claim on what benefits ARAP brings them. - I think building a working system based on previous successful architectures is totally fine. But I do agree that the authors shall conduct more study on this to validate their claim, eg, comparing the results of their correspondence network with the original PWC-net; replacing their correspondence network with PWC and compare the final error, etc. Otherwise its a bit hard to tell if the network is working as they claimed.

Correctness: - I'm a bit skeptical regarding the claim in L168-170. Since the optimization process is not "feature-based" [29], the impact of the network on the solver shall not be as effective as prior work. I thus wonder to what extent can the "weights" and "correspondences" improve the convergent iterations.

Clarity: - I think it will be great to mention explicitly that the camera parameters are already given a priori. Personally I'm aware of this, but I think its still good to make things clearer for the readers.

Relation to Prior Work: The literature review is in general well-written. I think it will be even better if the authors be more specific that they are not the first one to leverage neural networks to predict the weight of optimization terms.

Reproducibility: Yes

Additional Feedback: In general this paper provides a good pipeline for non-rigid tracking. The performance is clearly state-of-the-art. Implementation details are also clearly presented. The paper will be even better if more ablations are provided (see above). As this point, I would give this paper a poster rating.


Review 3

Summary and Contributions: This paper introduces a differentiable non-rigid tracking solver, which enables end-to-end learning of correspondence prediction and its weighting. The key idea is to backpropogate the gradient information from the deformation solver to the correspondence prediction network and the weighting network, making the networks output prediction that are beneficial for non-rigid tracking. Experiments show that the proposed method enables more robust tracking and reconstruction than state-of-the-art approaches.

Strengths: This paper proposes a novel method that embeds the non-rigid tracking solver into neural network training and enables gradient backpropogation from deformation solver to correspondence prediction. As far as I know, this paper is the first one to do so and hence the idea is quite novel.

Weaknesses: - Additional comparison: Although I understand the proposed method is a general method without relying on scene priors (such as human templates), I think it would be informative to add an additional comparison against other template-based non-rigid reconstruction methods such as DoubleFusion[31]. - Two additional questions that need to be clarified: 1) Why using an optical flow network for correspondence prediction? Since depth information is available, I think a scene flow network would be a more reasonable choice. 2) Why not defining the "graph loss" on the whole point cloud (because in this way you can regularize the node rotation)? ========================================================== I initially recommended acceptance of this paper because I like the idea of making the matching & tracking pipeline differentiable and end-to-end. But as R1 and R4 pointed out, this paper lacks sufficient technical novelty given that [15] has already proposed a differentiable optimization solver. Unfortunately, in the rebuttal the authors didn't fully clarify the novelty of their method and claimed that the "main contribution is self-supervised correspondence weighting" (LINE 37), which I found unsatisfactory. I believe this paper needs to discuss more on its novelty and how it differs from previous arts. Therefore, I would like to adjust my rating from 7 to 6 and encourage the author to further refine the paper and resubmit it to another venue.

Correctness: Yes

Clarity: Yes, the paper is well written and easy to follow.

Relation to Prior Work: Yes. It would be better if the author can discuss in detail how the proposed neural non-rigid tracking solver differs from RegNet[11] and [A] because these papers also propose to learn to solve rigid/non-rigid alignment. [A] Learning to Optimize Non-Rigid Tracking. arXiv, 2003.12230.

Reproducibility: Yes

Additional Feedback: Is it possible to provide the texture fusion results? Because the reconstruction results of objects with rich texture can better demonstrate the tracking accuracy improvements in a fine-grain scale.


Review 4

Summary and Contributions: This paper focuses on selecting confident correspondence from an offer-the-shelf optical flow method, and then align two rgbd images under non-rigid deformation. To this end, a differentiable solver is applied to make the pipeline differentiable, and the optical flow and weight network be learned specifically to this non-rigid rgbd alignment task.

Strengths: The main strength of the paper is that it breaks the non-rigid tracking into two stages: first is the optical flow estimation and correspondence weighting, i.e. the photometric alignment, and second the deformable camera pose estimation, i.e. the geometric alignment. The network is only applied in the photometric alignment stage and the geometric alignment is a white-box minimization, which makes the learning easier and more explainable.

Weaknesses: The main weakness of this paper is lacking insights and novel formulations. Since the target audiences will only be a small group of people within NeuralPS community, the novelty of this paper is limited and can not inspire broader audiences. More specifically: The photometric alignment directly applies an existing optical flow method, which is not customized for this problem. At least the authors should consider to include geometric features into the correlation computation of flow to fully utilize the geometric information from depth sensors. The weighting scheme of the correspondence itself is a worth studying topic but this paper did not fully explore this part. The minimization procedure is more like a direct and basic implementation of existing solvers. The objective function is common and easy to minimize, and the number of sampled points is relatively small and a LU decomposition is appliable.

Correctness: The claims and method are correct, so is the empirical methodology.

Clarity: The paper is well written and easy to follow.

Relation to Prior Work: The relation with previous works is well discussed.

Reproducibility: Yes

Additional Feedback: It is a good pipeline in terms of performance but lacks some insights and contributions as an inspiring paper. I would recommend the authors to shift their focus on how to design a better photometric alignment model specifically for non-rigid RGBD alignment and detach the correspondence from the specific differentiable solver, i.e., the differentiable solver is only for learning good correspondence estimation and weighting during training, once they are learned we can insert the correspondence estimation and weighting into any other non-rigid tracking framework and achieve a consistent improvement. --------------Post Rebuttal Update----------------- I maintain my rating after reading the rebuttal and comments from other reviewers. I keep the opinion that this paper has a solid pipeline but lacks sufficient novelty as a research paper, which is similar to another reviewer. However, I am happy to see an improved version of this paper in the next venues.

[Author Response · NeurIPS 2020]



We thank all the reviewers for their feedback, and are happy that our work was found to be "quite novel" (R3), "well written" (R4), "easy to follow" (R3), and supported by "promising results" (R1), that achieves "clearly state-of-the-art performance" (R2) while "significantly reducing the computational time [..] by 85 times compared to prior art [4]" (R2).

**Relation to [4] and [15] (R1, R2).** We propose to learn non-rigid tracking in an end-to-end fashion, where the solution of the non-rigid alignment informs correspondence prediction. Our differentiable formulation allows learning correspondence confidence in a completely self-supervised manner, similar to robust optimization. Such self-supervision of correspondence confidences results in improved tracking performance vs. directly supervising this task. In contrast, DeepDeform [4] learns sparse matches by individual heatmaps, resulting in a high compute cost and inferior performance. [15] has a different focus, as it learns a Preconditioned Conjugate Gradient to speed up solver convergence, using descriptors on nodes in a pixel-aligned graph. Our approach, which works on general graphs, learns to robustify dense correspondence prediction for non-rigid tracking by learning self-supervised correspondence confidences.

**Robustness of dense vs. sparse correspondences (R1).** We combine the advantages of both by learning a weight for each correspondence. This is inspired by robust optimization, and outperforms sparse correspondence works, e.g., [4].

**Learning through introspection (R1).** This is an interesting direction; in our approach, we aim to avoid direct reliance upon methods which maintain their own failure modes in very challenging deformation scenarios.

**Related self-supervised RGB tracking (R1) and camera parameters (R2).** We will discuss [Wang et al. 19], [Li et al. 19] and [Lai et al. 20] in the revised version, as well as clarify that camera parameters are given a priori.

**Varying keyframe rates (R1).** Tab. 1; performance remains stable even though frame-to-frame deformations increase.

Table 1: Deformation error (mm): more frames means lower sampling rate, i.e., more frame-to-frame motion.

| Keyframe density | Deformation error (mm) |
|---|---|
| DeepDeform (filtering w/ neighboring frames) | 31.52 |
| Ours: keyframe every 100 frames | 30.70 |
| Ours: keyframe every 75 frames | 29.68 |
| Ours: keyframe every 50 frames | **28.72** |

**Warp loss is superset of graph loss (R2, R3).** We agree that this is true in our particular case. We found it to be a more general notation to disentangle them (e.g., for scenarios where graph nodes are not sampled on the RGB-D frame).

**ARAP edge re-weighting (R2).** We sample nodes uniformly on the surface, thus, all edges have similar length ($7.13 \pm 1.38$ cm). Hence, edge re-weighting changes EPE 3D only marginally ($25.70$ mm w/o vs. $25.49$ mm w/).

**How does the graph size affect the method (R2)?** We sample graph nodes with $5$ cm node coverage, which fits well with our setup of $11$ GiB for training. Using coarser graphs, with $10$ cm and $15$ cm node coverage resulted in poorer performance: $5.49\%$ and $8.33\%$ higher EPE 3D, as well as $7.34\%$ and $28.46\%$ higher Graph Error 3D, respectively. In turn, the memory footprint on the GPU during training (with batch size 4) decreases with node coverage: $10\,513$ MiB, $6153$ MiB and $5931$ MiB for $5$ cm, $10$ cm and $15$ cm node coverage, respectively.

**Segmentation of foreground object (R2).** Our self-supervised learning of correspondence weighting can robustly handle outliers that arise from noisy masks (note that the used annotated masks are not perfect segmentations).

**# optimization steps (R2).** We empirically found 3 solver iterations to be the best compromise between performance and computational cost. Unrolling 3 solver iterations instead of 1 / 2, results in $24.9\%$ / $0.16\%$ lower EPE 3D and $22.7\%$ / $0.23\%$ lower Graph Error 3D; 4 iterations only improves slightly wrt 3 ($0.04\%$ lower EPE 3D, $0.03\%$ lower Graph Error 3D); more than 4 does not change performance notably. We will include this ablation in the revised version.

**Nearest-neighbor vs. bilinear depth sampling depth (R2).** We found bilinear sampling to perform better, with $5.8\%$ lower EPE 3D and $6.29\%$ lower Graph Error 3D compared to nearest-neighbor sampling.

**Optical flow vs. scene flow (R3, R4).** Our main contribution is self-supervised correspondence weighting with end-to-end non-rigid tracking, rather than the flow backbone. Thus, we leverage a state-of-the-art optical flow as our backbone (enabling significant pretraining); we believe that scene flow could also provide a useful backbone.

**Texture fusion / DoubleFusion results on mocap dataset (R3).** We will be happy to include these comparisons.

**Ablation on weighting scheme (R4).** This is indeed an exciting avenue that we would like to further explore in the future. Nevertheless, we believe our ablation on learning correspondence confidences, i.e., learning those in a completely self-supervised fashion (only possible thanks to our differentiable non-rigid tracker) vs. using supervision, can already be regarded as an exploration of the weighting scheme.

[Meta-Review · NeurIPS 2020]

This paper generated much discussion amongst the reviewers. The reviewers felt positively about the paper as providing a good combination of deep learning with optimization. The proposed method achieves state-of-the-art performance and reduced computation time, and the experiments are fairly extensive. The paper is generally well written and organized. The biggest weakness of the paper is the minimal technical novelty; reviewers felt that the proposed method is mostly a combination of methods from previous work, with minor modifications. Thus, the reviewers felt that the submission is more of a “systems” paper. There was much discussion about how systems papers should be treated at NeurIPS. Further, as a systems paper, reviewers felt that the paper is missing ablation analyses that can show the importance of each component of their system (e.g. comparison of the correspondences of the proposed system to the correspondences estimated by PWC-Net). As a minor point, reviewers also were interested in seeing more analysis on the performance of the proposed method on larger non-rigid motions, which can be simulated by dropping frames in the test set.